# Infection-induced type I interferons critically modulate the homeostasis and function of CD8+ naïve T cells

Mladen Jergović [1], Christopher P. Coplen[1], Jennifer L. Uhrlaub [1], David G. Besselsen[2], Shu Cheng[3], Megan J. Smithey[1,4] & Janko Nikolich-Žugich [1 ✉]

Naïve T (Tn) cells require two homeostatic signals for long-term survival: tonic T cell receptor:self-peptide–MHC contact and IL-7 stimulation. However, how microbial exposure impacts Tn homeostasis is still unclear. Here we show that infections can lead to the expansion of a subpopulation of long-lived, Ly6C+ CD8+ Tn cells with accelerated effector function. Mechanistically, mono-infection with West Nile virus transiently, and polymicrobial exposure persistently, enhances Ly6C expression selectively on CD5hiCD8+ cells, which in the case of polyinfection translates into a numerical CD8+ Tn cell increase in the lymph nodes. This conversion and expansion of Ly6C+ Tn cells depends on IFN-I, which upregulates MHC class I expression and enhances tonic TCR signaling in differentiating Tn cells. Moreover, for Ly6C+CD8+ Tn cells, IFN-I-mediated signals optimize their homing to secondary sites, extend their lifespan, and enhance their effector differentiation and antibacterial function, particularly for low-affinity clones. Our results thus uncover significant regulation of Tn homeostasis and function via infection-driven IFN-I, with potential implications for immunotherapy.

[1] Department of Immunobiology and the University of Arizona Center on Aging, University of Arizona College of Medicine, Tucson, AZ, USA. [2] University Animal Care, University of Arizona, Tucson, AZ, USA. [3] Department of Medicine, University of Arizona College of Medicine, Tucson, AZ, USA. [4]Present address: Vir, Inc., San Francisco, CA, USA. ✉email: nikolich@email.arizona.edu

Naïve T cells (Tn) are mature post-thymic T cells that reside in secondary lymphoid organs/tissues, awaiting encounter with their cognate antigen. In mice, Tn cells express L-selectin (CD62L) and CCR7 but not the memory/activation markers CD44, CD25, or CD49d[1]. Adequate numbers and diversity of Tn cells are essential for immune defense against newly encountered pathogens[2]. T cell homeostasis maintains the numbers and diversity of the Tn pool[3]. Seminal findings of Suhr, Sprent, and colleagues have shown that Tn cells require tonic, subthreshold T cell receptor (TCR) stimulation by self-peptide:MHC (s-pMHC), followed by binding of lymph node stroma-produced IL-7 to the IL-7 receptor (IL-7R), for survival and maintenance, as well as for homeostatic proliferation[4]. These and almost all other studies on Tn homeostasis were done in inbred mice housed under specific pathogen-free (SPF) or even germ-free or antigen-free[5] conditions, in the almost complete absence of infection and inflammation. A recent landmark study[6] compared immune homeostasis in wild-caught mice, mice from pet stores (PS), and inbred, laboratory SPF mice. Authors found that SPF mice exhibited T cell subset distribution and activation patterns similar to that of human neonates, whereas the same parameters in wild-caught, pet-store, or inbred SPF mice co-housed with PS mice, faithfully approximated the distribution of T cell subsets (with low levels of naïve and high levels of memory cells), T cell subset activation and responses to infection seen in adult humans.

We previously described a subset of phenotypically naive human CD8+ T cells with memory characteristics[7] that rapidly secreted multiple cytokines in response to persistent infections. In our hands, no similar population could be found in SPF mice. Given the powerful impact of non-SPF conditions on the murine immune system, we here investigated the phenotype and function of CD8+ Tn cells in C57BL/6 mice following viral or bacterial

mono-infection, as well as following co-housing exposure to pet-store animals (CH in the text).

Here we show that infections and inflammation have a long-lasting antigen-nonspecific (bystander) effect upon CD8+ Tn cell homeostasis, dominated by an expansion of CD8+ Tn cells expressing a memory marker Ly6C. In CH animals, the expansion of CD8+ Tn Ly6C+ is driven by type I interferons (IFN-I) and dependent upon tonic TCR signaling. Compared to their Ly6C− counterparts, Ly6C+CD8+ Tn cells are preferentially home to lymph nodes and display enhanced homeostatic and effector properties. Overall, our results demonstrate a novel and powerful role of infection-mediated IFN-I in CD8+ Tn homeostasis.

## Results

**Unbiased clustering identifies upregulation of Ly6C on the surface of CD8+ Tn cells following WNV infection.** To examine the impact of infection upon the maintenance of Tn cells, we infected C57BL/6 (B6) mice with the West Nile virus strain 385-99[8]. Flow cytometric analysis was performed on day 7 (d7, peak of the immune response) in an unbiased manner to reveal potential changes in the circulating T cell pool, using a 15-marker flow cytometric panel, including commonly used memory and activation markers (Table S1). We have used the Flowsom visualization algorithm[9], an unsupervised technique for clustering and dimensionality reduction, to visualize changes among CD8+T cell subsets following WNV infection. Flow-SOM software visualizes flow cytometric data as a minimal spanning tree result very similar to more commonly used SPADE software but with greater computing speed. Flowsom identified the standard subsets of CD8+T cells, including the effector/effector memory ((EM) CD44hiCD62L−), central memory ((CM) CD44hiCD62L+CD49d+), virtual memory ((VM) CD44hiCD62L+D49d−) and naïve ((N)

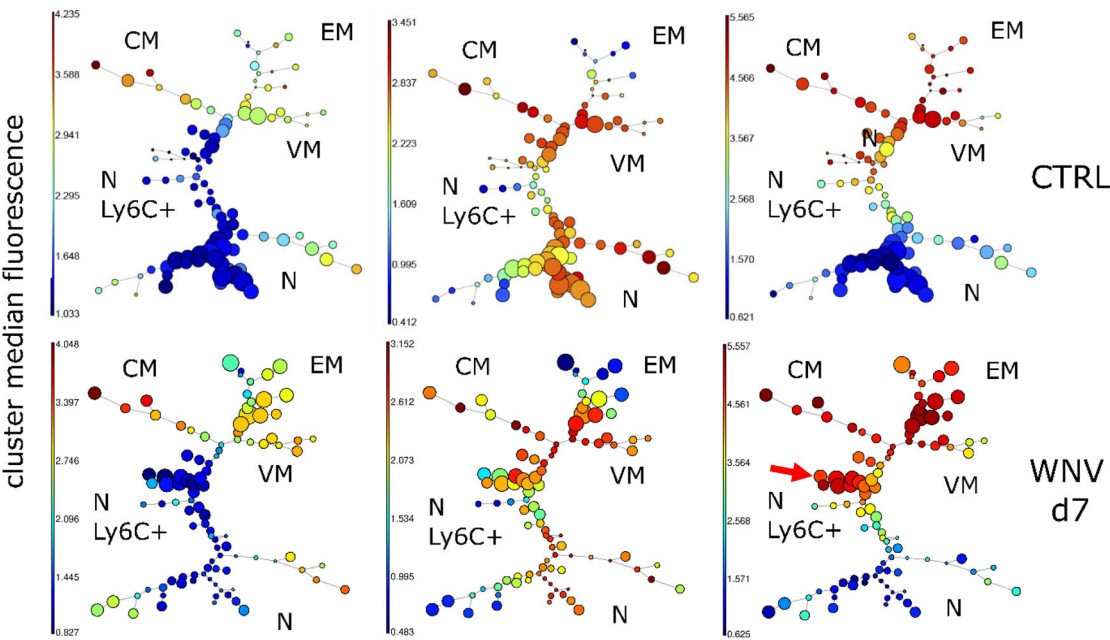

**Fig. 1 Unbiased clustering identifies upregulation of Ly6C on the surface of CD8+ Tn cells following WNV infection.** C57BL/6 mice (N = 9) were infected with 1000 PFU WNV virus by footpad injection. WNV infected and uninfected control mice (N = 9) were bled retro-orbitally at days 7 post-infection and stained with a 15 marker flow cytometry panel. FCS files were uploaded in Cytobank, cloud-based clustering software, CD3+ live cells were gated prior to clustering. Flowsom visualization algorithm identified the standard subsets of CD8+T cells effector/effector memory ((EM) CD44hiCD62L−), central memory ((CM) CD44hiCD62L+CD49d+), virtual memory ((VM) CD44hiCD62L+CD49d−) and naïve ((N) CD44loCD62L+CD49d−). Tn was divided into two subsets based on Ly6C expression, Ly6C+ subset (indicated by red arrow) was enlarged at d7 of WNV infection. Data is from one experiment (with N = 9 mice per group).

$CD44^{lo}CD62L^{+}CD49d^{-}$) cells (Fig. 1). However, the naïve T cells were divided into two distinct subsets, divided by the expression of Ly6C, with the $Ly6C^{+}$ cells unexpectedly and markedly increased in WNV-infected mice (Fig. 1, arrow). Therefore, unbiased clustering identified the upregulation of Ly6C on $CD8^{+}$Tn cells as one of the most pronounced changes in circulating T cells during WNV infection.

**Ly6C is upregulated on phenotypically naïve $CD8^{+}$ T cells transiently during acute viral and bacterial infection and permanently under 'non-SPF' conditions.** Next, we analyzed the kinetics of Ly6C upregulation. Flow cytometric strategy for identifying $Ly6C^{+}CD8^{+}$ Tn cells is shown in Fig. 2A, (depicting day 7 post-infection with WNV) and was used for all analysis, unless otherwise indicated. WNV-infected mice displayed more than the twofold higher expansion of the $Ly6C^{+}CD8^{+}$ Tn cells by day 3 p.i. (Fig. 2B) compared to day 0 of infection. However, the upregulation of Ly6C on $CD8^{+}$ Tn was transient. Ly6C expression declined by d16 and returned to baseline by d28 p.i. Interestingly, virtually all $CXCR3^{+}CD8^{+}$ Tn cells were contained within the $Ly6C^{+}$ subpopulation (Fig. 2A), making them phenotypically similar to human $T_{MNP}$ cells which displayed higher CXCR3 expression[7]. The proportion of double-positive $Ly6C^{+}CXCR3^{+}$ cells was also higher on d7 post-WNV infection (Fig. S1A) but this increase was lower than the ~2 fold increase in the total $Ly6C^{+}$ population. To examine whether the Ly6C upregulation on $CD8^{+}$ Tn cells was specific to viral infections, we repeated our experiments using the bacterium *Listeria monocytogenes* (Lm). In Lm-infected mice we found similar transient increased expression of Ly6C on $CD8^{+}$ Tn cells, peaking on day 3 but returning to baseline somewhat earlier than during the WNV infection (Fig. 2C).

We next measured Ly6C expression longitudinally in mice co-housed (CH) with pet store mice. As previously described[6], we found that cohousing of laboratory mice with PS mice resulted in transmission of multiple pathogens (Table S2). To examine the systemic effects of these infections, we performed measurements of multiple cytokines in secondary lymphoid tissue homogenates on day 60 after cohousing. Multiple cytokines were elevated as depicted in the heatmap in Fig. 2D. These results indicate that transmission of multiple pathogens to B6 mice resulted in prolonged stimulation of the immune system, evidenced by the elevation of multiple cytokines even after 60 days of continuous cohousing. Under such conditions, upregulation of Ly6C on $CD8^{+}$ Tn was permanent and did not return to baseline even after 300 days following CH (Fig. 2E). Female mice were used for all cohousing experiments and their basal Ly6C expression on $CD8^{+}$ Tn cells was lower than their SPF male counterparts (Fig. S1B). Only a slight upregulation of Ly6C was observed on CD4 T cells in CH mice (Fig. S1C). Similarly, the number of double-positive $Ly6C^{+}CXCR3^{+}$ naïve $CD8^{+}$s was also increased long term (Fig. S1D). Thus, the long-lasting increase in Ly6C expression in 'non-SPF' mice might be maintained by persistent/chronic infections and/or by continuous exchange of pathogens between infected individuals, reflective of a situation found in the naturally-dwelling mammalian communities. While transient bystander effects of inflammatory cytokines on memory T cells have previously been reported[10,11] our results indicate that chronic infections and inflammation can powerfully remodel even the Tn pool over long periods of time.

**$Ly6C^{+}$ $CD8^{+}$ Tn cells display rapid and enhanced effector function in vitro.** Previously we have shown, in human subjects, an increase in $CD8^{+}$ T memory cells with naïve phenotype ($T_{MNP}$) which were specific for persistent microbial pathogens[7]. Such cells

exhibited signs of basal activation and displayed rapid effector function upon polyclonal stimulation. To investigate whether the $Ly6C^{+}CD8^{+}$ Tn cells in mice cells exhibit enhanced functional response, we sorted $Ly6C \pm CD3^{+}CD8^{+}CD62L^{+}CD44^{lo}$ cells from secondary lymphoid tissues of SPF and CH mice (Fig. 3A, complete gating strategy in Fig. S7A) and stimulated them with α-CD3/α-CD28 beads and phorbol-myristate acetate (PMA) and the calcium ionophore ionomycin. Following 24 h of stimulation with α-CD3/α-CD28 beads, GzB expression was two-fold higher in $Ly6C^{+}$ over $Ly6C^{-}$ cells from CH mice while both cell types from SPF mice produced very little GzB (Fig. 3B, left panel). After 36 h, $Ly6C^{+}$ cells from both groups of mice produced more GzB than their $Ly6C^{-}$ counterpart although the difference was now less than two-fold (Fig. 3B, right panel). As expected, of Tn cells short (6 h) PMA/ionomycin stimulation resulted in low IFN-γ production (~1%); nonetheless, twice as many $Ly6C^{+}$ cells produced IFN-γ in cells from both SPF and CH mice (Fig. 3D). Production of TNF-α was much higher, again with the same pattern of increased production by $Ly6C^{+}$ cells in both groups of mice (Fig. 3E). This suggested that $Ly6C^{+}$ cells may be at a higher basal activation state even in SPF mice, prompting us to examine phosphorylation of the Erk kinase, which integrates several signaling pathways in T cells[12]. We found that pErk levels at basal state and after 20 min of polyclonal stimulation with α-CD3/α-CD28 beads, showed a small but significant increase in $Ly6C^{+}$ cells (Fig. 3F).

This suggested that $Ly6C^{+}$ cells might have been receiving increased TCR and/or cytokine signals, leading us to measure levels of phosphorylated ZAP-70/SYK. Upon TCR engagement, phosphorylation of ZAP-70 Y319 by Lck is necessary for T cell receptor (TCR)-dependent association of ZAP-70 with Lck and downstream signaling[13]. $Ly6C^{+}$ cells at both the basal and activated states expressed similarly small, but reproducibly and significantly higher levels of pZAP-70/SYK (Fig. 3G), further supporting the possibility that these cells are receiving TCR-mediated signals. Given that overall numbers of $Ly6C^{+}$ cells in CH, mice were twofold higher, these results indicated that in CH mice most of the $CD8^{+}$ Tn pool was in a state of higher basal activation, affording them higher production of effector molecules.

**Microbial colonization increases Ly6C expression on naïve T cells in a bystander manner.** Since both Ly6C and CXCR3 have been reported to be expressed primarily on effector and memory T cells[14] it was possible that our $Ly6C^{+}$ population was contaminated by recently activated effector T (Te) cells. We examined the expression of multiple memory and activation markers such as CD49d, CD25, KLRG1, and CD69 on phenotypically naïve $CD8^{+}$ T cells from cohoused mice. All of these markers are upregulated upon activation via TCR, with CD69 exhibiting the earliest upregulation[15]. We found that effector/memory $CD44^{hi}$ cells in CH mice expressed KLRG1, CD49d, and CD69. By contrast, $CD8^{+}$ Tn cells failed to express any of these markers regardless of Ly6C expression (Fig. 4A), implying that this population contains very few cells recently stimulated by antigen. Next, using fluorescence-minus-one control, we divided the $CD44^{lo}$ $CD62L^{hi}$ cells into the CD44 negative and CD44 intermediate subpopulations and compared their Ly6C expression to that on $CD44^{hi}CD62L^{hi}$ Tcm cells. $CD44^{lo}CD62L^{hi}$ cells doubled their expression of Ly6C upon cohousing (Fig. 4B). While $CD44^{int}CD62L^{hi}$ cells contained a higher fraction of $Ly6C^{+}$ cells at baseline, cohousing also upregulated their expression of Ly6C; no such upregulation was seen on $CD44^{hi}$ CM cells. One could argue that the fraction of $CD8^{+}CD44^{int}CD62L^{hi}$ cells, commonly included in the Tn gate in the field, as well as in most of our experiments, represent recently activated memory-like cells that could obscure our results. However, the above results argue

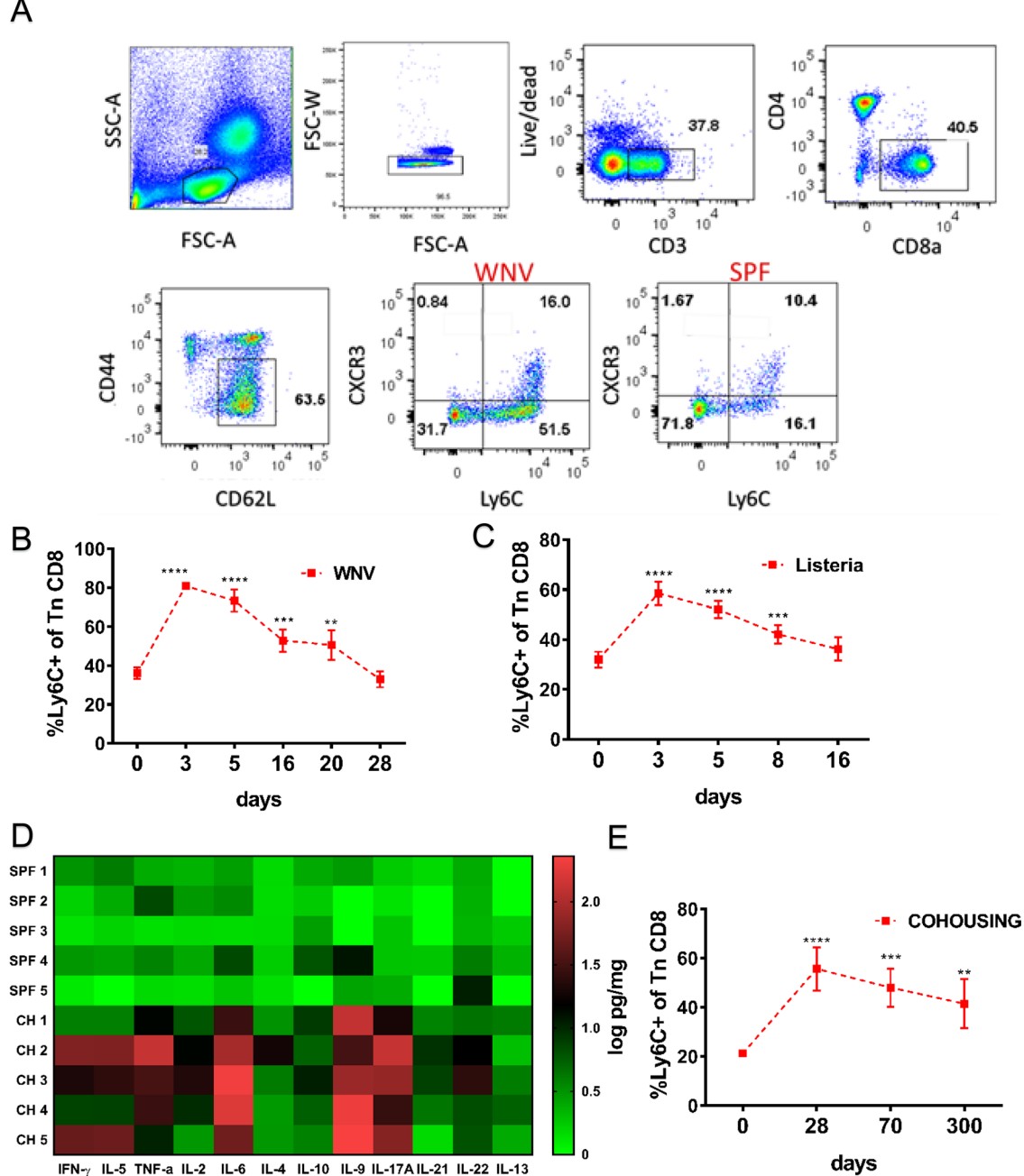

**Fig. 2 Naïve CD8$^+$ T cells upregulate Ly6C transiently during acute viral and bacterial infection and permanently under 'non-SPF' conditions. A** Flow cytometry gating strategy to analyze the expression of Ly6C and CXC3 on naïve (CD44$^{lo}$CD62L$^{hi}$) CD8$^+$ T cells Briefly, live CD3$^+$CD8$^+$ cells were gated with the standard phenotypic definition for Tn cells (CD62L$^{hi}$CD44$^{lo}$) and then divided into the positive and negative population by Ly6C expression. **B** C57BL/6 mice were infected with 1000 PFU WNV virus by footpad injection. Mice were bled retro-orbitally at days 3, 5, 16, 20, and 28 post-infection. Blood was hypotonicially lysed and surface stained with antibodies for flow cytometric analysis. ($n = 5$ mice per group, data presented as mean ± sd, **$p = 0.0049$, ***$p < 0.001$, ****$p < 0.0001$) **C** Mice were inoculated intravenously (i.v.) with $10^4$ CFU Listeria monocytogenes. Mice were bled retro-orbitally at days 3, 5, 8, and 16 p.i. The proportion of Ly6C$^+$ positive CD44loCD62Lhi cells were analyzed by flow cytometry ($n = 5$ mice per group, data presented as mean ± sd, ***$p < 0.001$, ****$p < 0.0001$). **D** Expression of various inflammatory cytokines was measured in homogenates of secondary lymphoid tissue of cohoused and SPF control mice by Legendplex flow cytometry multiplex immunoassay ($n = 5$ mice per group) (**E**) cohoused mice were bled periodically and proportion of Ly6C$^+$ positive naïve CD8$^+$s was measured by flow cytometry ($n = 5$ mice per group, data presented as mean ± sd, **$p = 0.0022$, ***$p < 0.001$, ****$p < 0.0001$). **A–E** Data are representative of two independent experiments (with $n = 5$ mice per group) (*$p < 0.05$, **$p < 0.01$, ***$p < 0.001$, ****$p < 0.0001$). **B–E** unpaired one-way ANOVA with Dunett's correction for multiple comparisons.

against that, because these cells did not exhibit upregulation of antigen-induced activation markers (Fig. 4A) and, while these cells expressed a higher frequency of Ly6C$^+$, they were still able to upregulate it further in response to cohousing (Fig. 4B).

Previously we have shown in a parabiosis model that naïve CD44$^{lo}$CD62L$^{hi}$ cells freely circulate and equilibrate between parabiont host[16]. Thus, it was not surprising that Ly6C upregulation on Tn CD8$^+$s was detected at similar levels in

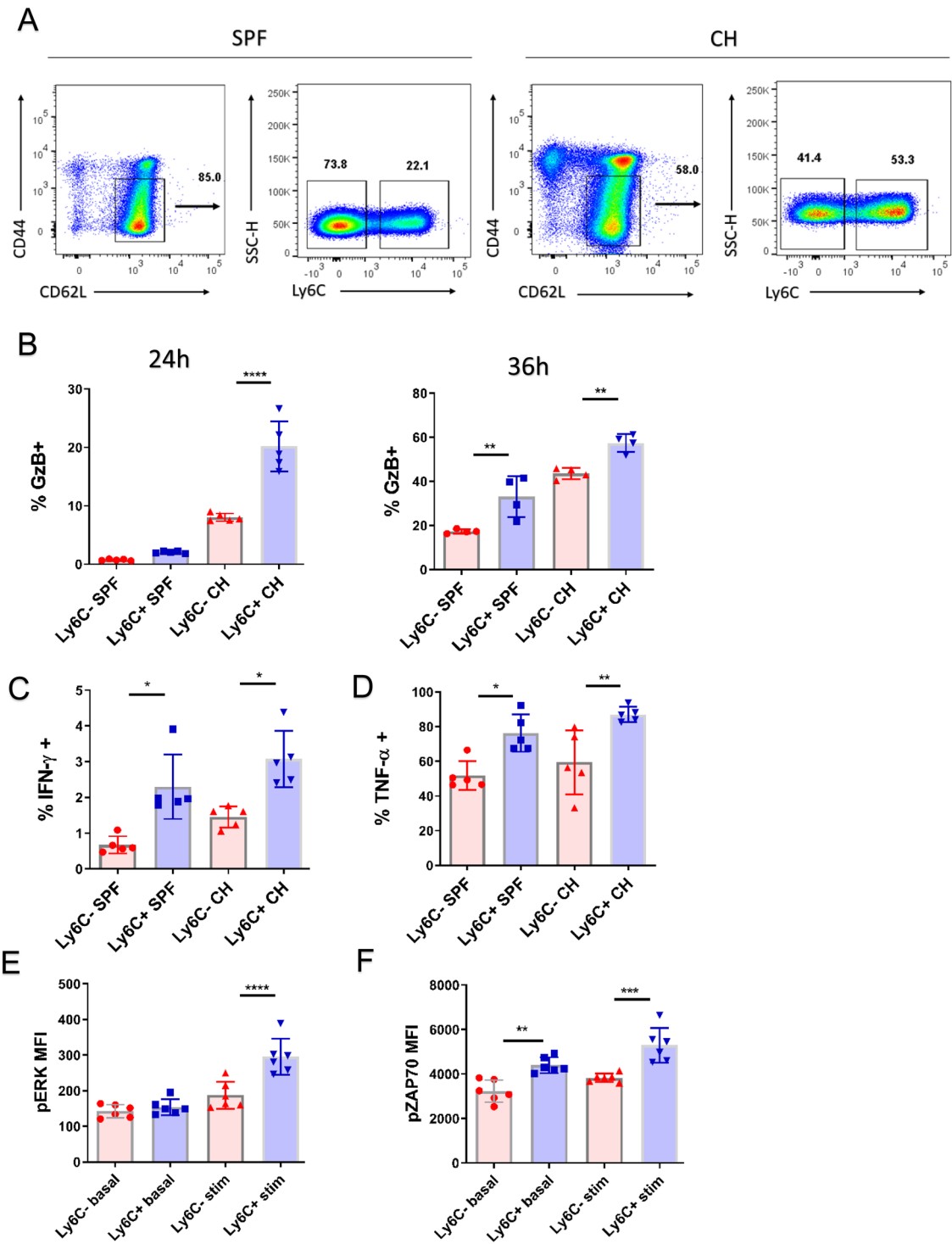

blood, spleen, and lymph nodes (Fig. 4C). To further confirm that increase in Ly6C expression is of bystander type and not a result of TCR cross-reactivity, we treated SPF mice with poly(I:C), an innate immune agonist known to induce multiple inflammatory cytokines[17]. Treated mice upregulated Ly6C on Tn (Fig. S2A) to a similar extent previously observed in WNV infection. To confirm that Ly6C upregulation is not dependent on cognate antigen we have transferred CFSE labeled Tn CD8+s from OT-1 mice into cohoused mice. After 14 days transferred cells have largely remained CD44$^{lo}$CD62L$^{hi}$ and have not diluted out CFSE; however, >70% of the undivided cells upregulated Ly6C (Fig. 4D). Jointly, these results confirm that Ly6C upregulation is a bystander effect of inflammation on CD8+ Tn cells and not a result of antigenic stimulation or contamination with T effector or memory cells.

To examine in detail the differentiation status of Ly6C+CD8+ Tn cells, we performed extensive FCM analysis of multiple surface proteins (chemokines, chemokine receptors, adhesion molecules, cytokine receptors, and costimulatory molecules) involved in T cell homeostasis and maturation. One pronounced change observed on all CD8+ Tn cells in co-housed mice (CH) was the upregulation of stem cell antigen 1 (SCA-1) (Fig. 4E). SCA-1, which also belongs to the Ly-6 superfamily, is expressed on memory T cells and gets upregulated on almost all T cells

**Fig. 3 Ly6C$^+$ CD8$^+$ Tn cells display rapid and enhanced effector function in vitro. A** CD8 T cells were enriched by magnetic sorting from secondary lymphoid tissues (spleen and pool of brachial, inguinal, and cervical lymph nodes) of SPF and CH mice. Ly6C$^+$ and Ly6C$^-$ CD44$^{lo}$CD62L$^{hi}$ cells were then sorted by flow cytometric sorting. Forty thousand sorted cells were plated in round-bottom plates and simulated α-CD3/α-CD28 beads in 1:1 cell to bead ratio for 24 h or 36 h and with phorbol-myristate acetate (PMA) and ionomycin for 6 h in presence of brefeldin A. **B** Expression of GzB was measured on live cells by intracellular flow cytometric staining following 24 h (left panel, $n = 5$ mice per group, data presented as mean ± sd, left to right **$p = 0.0021$, **$p = 0.0055$) or 36 h stimulation with α-CD3/α-CD28 beads (right panel, $n = 4$ mice per group, data presented as mean ± sd) (**C**) IFN-γ expression was measured on live cells by intracellular flow cytometric staining ($n = 5$ mice per group, data presented as mean ± sd, left to right *$p = 0.0184$, *$p = 0.0479$), as well as expression of (**D**) TNF-α. ($n = 5$ mice per group, data presented as mean ± sd, *$p = 0.0131$, **$p = 0.0055$). **E** Phosphorylation of Erk (p-Erk) was measured by phosflow on unstimulated sorted cells or stimulated for 20 min with an α-CD3/α-CD28 bead ($n = 6$ mice, data presented as mean ± sd, ****$p < 0.0001$), as well as (**F**) phosphorylation of ZAP-70 Y319 ($n = 6$ mice, data presented as mean ± sd, **$p = 0.0014$, ***$p < 0.001$). **A, C, D** Data are representative of two experiments (with $n = 5$ mice per group). **E, F** Data are representative of two experiments (with $n = 6$ mice) (*$p < 0.05$, **$p < 0.01$, ***$p < 0.001$, ****$p < 0.0001$). **A, B, D, E, F** one-way ANOVA with Sidak post-hoc correction for multiple comparisons: C-one way Kruskal-Wallis test with Dunn's correction for multiple comparisons.

during viral infection[18]. However, SCA-1 levels were increased in CD8$^+$ Tn cells in CH mice regardless of the Ly6C expression. Another surface marker, CD5 was highly expressed on Ly6C$^+$ cells from both SPF and CH mice (Fig. 4E). CD5 is a negative regulator of T cell receptor (TCR)-mediated signaling[19] and its expression on T cells correlate with TCR affinity so that clonotypes with higher affinity for self p-MHC exhibit higher CD5. Increased levels of CD5 on Ly6C$^+$ cells together with increased pZAP-70/SYK (Fig. 3D) suggested that tonic TCR signals may play a role in the maintenance of this population. We further measured intracellular levels of multiple molecules involved in T cell homeostasis and activation/memory maintenance. Ly6C$^+$ CD8$^+$ Tn cells from both CH and SPF mice exhibited significantly higher expression of Eomesodermin (Eomes) (Fig. 4F) and BCL-2 (Fig. 4G), as well as slightly higher levels of IRF-4 and MCL-1 (Fig. S2B) with no expression/difference in Tbet, BCL-6 and BLIMP-1 levels (Fig. S2C). Eomes is a key transcription factor involved in the generation of effector and memory T cells[20] and in the maintenance of the memory T cell pool[21]. BCL-2 is an anti-apoptotic factor that is positively regulated by homeostatic cytokines such as IL-7[22] and the expression of which correlates with prolonged cell survival. We conclude that CD8$^+$ Tn cells from cohoused mice are phenotypically and functionally altered. Their dominant phenotype, as observed by flow cytometry was Ly-6C$^+$CD62L$^{hi}$CD44$^{lo}$ Eomes$^{hi}$BCL-2$^{hi}$CD5$^{hi}$SCA1$^{hi}$.

**RNA-seq reveals Ly6C$^+$ cells as a transcriptionally distinct subset of CD8$^+$ Tn cells.** We next performed RNA-seq on CD8$^+$ Tn cells from both SPF and CH mice sorted by Ly6C expression to define Ly6C-dependent transcriptomic differences. When analyzed independently, principal component analysis (PCA) of global gene expression of Ly6C$^+$ vs. Ly6C$^-$ Tn cells yielded robust clustering of samples in SPF mice (51% of the variance, PC1), with markedly more modest clustering in CH mice (36% of the variance, PC1). When analyzed together, global gene expression of all samples affirmed a strong transcriptional difference between SPF and CH mice, associated with the first principal component (PC1), accounting for 41% of the total variance (Fig. 5A). Differential gene expression analysis between the Ly6C$^+$ and Ly6C$^-$ cells identified 154 (SPF) and 46 (CH) significantly [false discovery rate (FDR) < 0.05] downregulated genes, and 220 (SPF) and 50 (CH) upregulated genes (Fig. 5B). Not surprisingly, *Ly6C1* and *Ly6C2* exhibited the most significant upregulation, and the results confirmed our previous observations made at the protein level, such as the increased expression of Eomes and CXCR3 mRNA in Ly6C$^+$ CD8 Tn cells (Fig. 5B–D, S3A). Fifty of the 374 significantly differentially expressed genes from SPF mice overlapped significantly ($P < 2.2e-16$, Fisher's exact test) with those 96 genes from CH mice (Fig. 5C). Gene

ontology (GO) and KEGG enrichment analysis identified multiple immune-specific pathways ($q < 0.1$) including "chemokine signaling pathway" and "viral protein interaction with cytokine and cytokine receptor" (Fig. S3B, C). Despite the differences between SPF and CH mice (Fig. 5A), hierarchical clustering of the 50 common gene expression matrix showed that the Ly6C$^+$ cell cluster in both SPF and CH mice (dark green) was separate from their Ly6C$^-$ counterpart (light green) (Fig. 5D). These data support a model whereby, at the genome-wide scale, a distinct subset of immune function-related genes is transcribed to convey enhanced effector function to the Ly6C$^+$ CD8 Tn cells in both SPF or CH mice, consistent with data from Fig. 3.

**Ly6C upregulation is mediated by IFN-I and involves tonic TCR recognition of pMHC.** Given that Ly6C expression on CD8$^+$ Tn cells is induced in a bystander manner, a mode of action consistent with the influence of a soluble mediator, we have examined multiple inflammatory and homeostatic cytokines for their ability to upregulate Ly6C$^+$ on CD8$^+$ Tn cells. We have cultured CD8$^+$ Tn cells for 24 h with 100 ng/ml of each cytokine. Only IFN-I (both IFN-α and IFN-β), we're able to induce Ly6C expression (Fig. 6A). This was consistent with Ly6C$^+$ cells expressing high levels of Eomes, as Eomes expression in CD8$^+$ T cells has been shown to be regulated by IFN-I[23]. Recently it has been reported that IL-27 can induce moderate Ly6C expression on Tn cells after 72 h after culture[24], yet we observed no such effect after 24 h. To confirm the role of IFN-I in vivo, we have measured Ly6C expression in mice lacking IFN-I receptor (Ifnar1$^{-/-}$ mice). Ly6C$^+$ population was almost entirely missing in Ifnar1$^{-/-}$ mice as only ~5% CD8$^+$ Tn cells expressed Ly6C (Fig. 6B). While Ifnar1$^{-/-}$ mice are known to be highly susceptible to most viral diseases, they have been shown to be resistant to Listeria infection[25]. Therefore, to assess the ability of CD8$^+$ Tn cells to upregulate Ly6C during infection in the absence of IFN-I signaling we have infected Ifnar1$^{-/-}$ mice. We found that while Ifnar1$^{-/-}$ mice were able to slightly upregulate Ly6C expression on CD8$^+$ Tn cells on d3 post-infection, the fraction of Ly6C$^+$ cells in these mice was at least threefold lower than in Lm-infected wt mice (Fig. 6B). Therefore, we conclude that IFN-I is the main and necessary soluble signal for induction of Ly6C expression on Tn cells. We then measured levels of type I IFNs in serum of SPF and CH mice and observed long term (6 mo) upregulation of type I IFNs in continuously CH mice (Fig. S4A) explaining the previously observed long term upregulation of Ly6C in these animals (Fig. 2F).

In light of the critical role of IFN-I, the increased pZAP-70/SYK and CD5 levels could signify an additional role for TCR-mediated signals in the upregulation of Ly6C. Specifically, IFN-I is a well-known inducer of MHC molecules, potentially providing an increase in TCR ligands for homeostatic, tonic TCR stimulation on CD8$^+$ Tn cells. To investigate the potential role of sp-MHC affinity and tonic TCR signals in the upregulation of

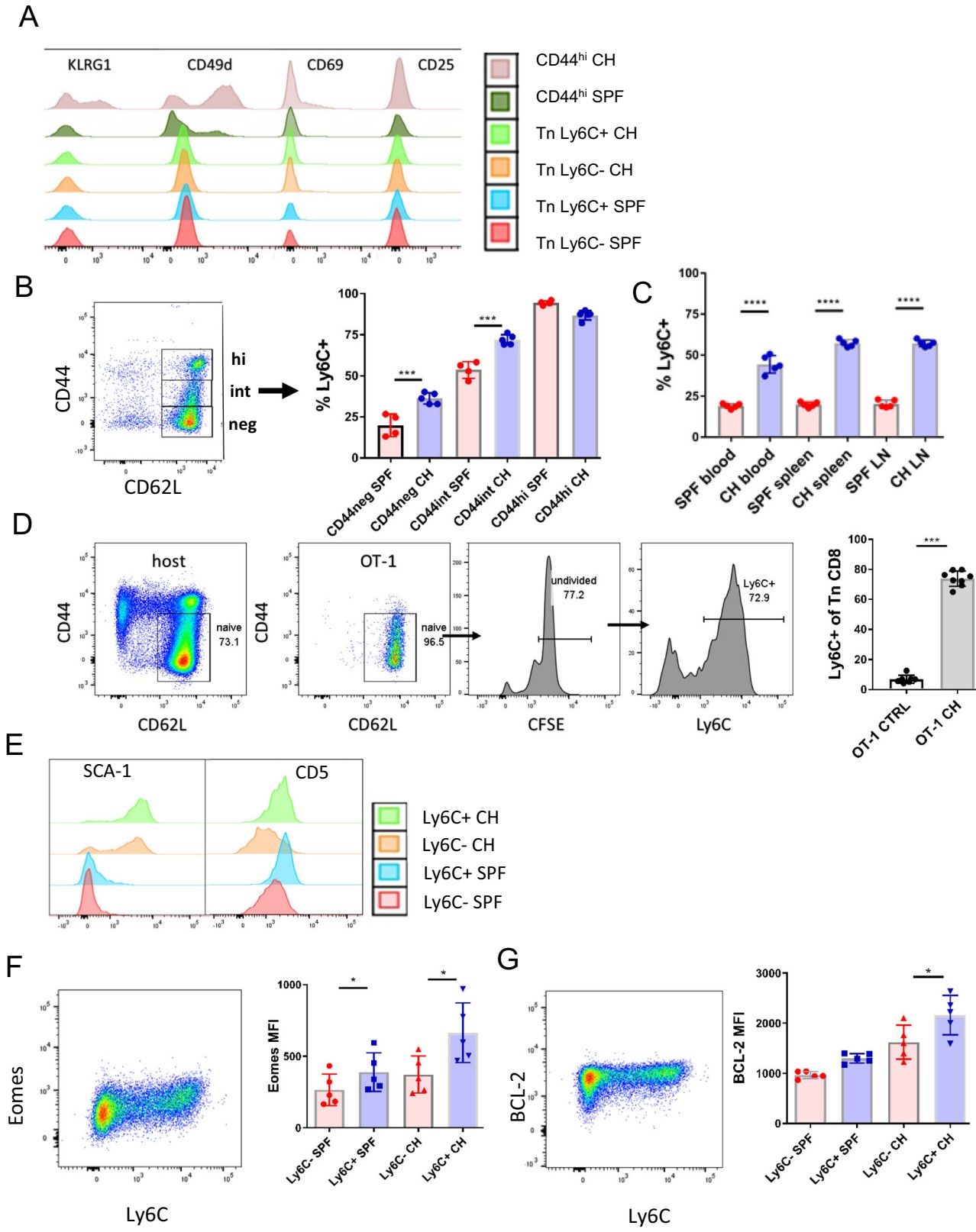

Ly6C, we sorted the top and bottom 20% of CD8$^+$ Tn cells by CD5 expression (Fig. 6C, left panel). Previously it was reported that CD5$^{hi}$ Tn cells had some similarities to memory CD8$^+$ T cells[26], including higher CD44, CXCR3, and Eomes expression. Here we tested the ability of CD5$^{hi}$ and CD5$^{lo}$ cells to upregulate Ly6C in response to IFN-I. We found that most of the CD5$^{hi}$ cells upregulated Ly6C robustly, whereas very few CD5$^{lo}$ cells did (Fig. 6C) when stimulated with low (10 ng/ml) IFN-α. By contrast, both populations were able to upregulate SCA-1 (Fig. 6C, right panel). With higher IFN-α concentrations of 100 ng/ml, a subset of ~40% of CD5$^{lo}$ cells was also able to upregulate Ly6C, which was still less compared to >80% of the CD5$^{hi}$ cells

**Fig. 4 Microbial colonization increases Ly6C expression on CD8$^+$ Tn cells in a bystander manner. A** Female C57BL/6 mice were cohoused with outbred, wild-type pet shop mice in large rat cages, separated by a perforated barrier. After >60 days of cohousing expression of Cd49d, CD69, KLRG1, CD27, and CD25 was analyzed by flow cytometry on subpopulations of Tn CD8$^+$ T cells from lymph nodes of CH mice and SPF controls (Color scheme: red-Tn Ly6C$^-$ from SPF mice; blue-Tn Ly6C$^+$ from SPF mice; orange-Tn Ly6C$^-$ from CH mice; light green-Tn Ly6C$^+$ from CH mice; dark green-CD44$^{hi}$ from SPF mice; brown-CD44$^{hi}$ from CH mice). **B** Upregulation of Ly6C was upregulated on central memory (CM) cytotoxic T cells (CD8$^+$CD44$^{hi}$CD62L$^{hi}$), CD44$^{neg}$CD62L$^{hi}$ and CD44$^{int}$CD62L$^{hi}$ cells ($n = 4$ SPF mice, $n = 5$ CH mice per group, data presented as mean $\pm$ sd, ***$p < 0.001$). **C** Ly6C was upregulated on Tn CD8$^+$ cells in the blood, lymph nodes, and spleen ($n = 5$ mice per group, data presented as mean $\pm$ sd, ****$p < 0.0001$). **D** Tn CD8$^+$ cells were magnetically enriched, CFSE labeled and transferred into CH mice for 14 days ($n = 7$ ctrl mice, $n = 8$ CH mice, data presented as mean $\pm$ sd, ***$p = 0.0003$). **E** Expression of CD5 and SCA-1 on Ly6C $\pm$ CD8$^+$ Tn from SPF and CH (Color scheme: red-Tn Ly6C$^-$ from SPF mice; blue-Tn Ly6C$^+$ from SPF mice; orange-Tn Ly6C$^-$ from CH mice; light green-Tn Ly6C$^+$ from CH mice). **F** Ly6C $\pm$ CD8$^+$ Tn from SPF and CH ($n = 5$ mice per group, data presented as mean $\pm$ sd, left to right *$p = 0.0481$, *$p = 0.0336$) were stained intracellularly for expression of Eomes and (**G**) BCL-2 ($n = 5$ mice per group, data presented as mean $\pm$ sd, *$p = 0.0168$). **A**, **B**, **C**, **E**, **F**, and **G** data are representative of two experiments (with $n = 5$ mice per group). **D** Data are pooled from two experiments (total $N = 7$ mice per group (*$p < 0.05$, **$p < 0.01$, ***$p < 0.001$, ****$p < 0.0001$). **D** Two-tailed Mann-Whitney U-test; **A**, **B**, **C**, **E**, **F**, and **G**) one-way ANOVA, with Sidak post hoc correction.

(Fig. 6D). This result strongly suggested that tonic TCR signaling may be necessary as an additional signal to upregulate Ly6C following IFN-I stimulation. Because in our experiment CD8$^+$ Tn cells were cultured alone, we inferred that this signal could only be coming from neighboring T cells. To address this issue, we stimulated highly purified CD8$^+$ Tn cells with IFN-α in the presence of an MHC-I blocking antibody. Under such conditions, the IFN-α mediated upregulation of Ly6C was completely abrogated (Fig. 6E, left), while the expression of SCA-1 was significantly reduced (Fig. 6E, right). Therefore, in addition to IFN-I signaling, tonic MHC-dependent TCR signaling was necessary to induce Ly6C on CD8$^+$ Tn cells. To further investigate this relationship in vivo, we have treated Nur77$^{GFP}$ transgenic reporter mice with 0.5 μg/mouse IFN-α. Nur77 mice express GFP from the immediate early gene Nr4a1 (Nur77) locus which is upregulated by TCR stimulation, but not by inflammatory stimuli[27]. Upon treatment with IFN-α, Ly6C was upregulated on CD8$^+$ Tn cells (Fig. 6F, left) to a similar extent previously observed in WNV infection (Fig. 2A). The Nur77$^{GFP}$ signal was increased on all CD8$^+$ Tn cells by IFN-α treatment (Fig. 6F, right) but MFI was higher on Ly6C$^+$ cells (Fig. 6G, left). This increase in GFP signal was, as expected, much lower than the one elicited by stimulation of the same cells by cognate antigen (Nur77 OT-1 stimulated with SIINFEKL peptide, Fig. 6G, right), consistent with differences in signal intensities for antigenic vs. tonic subthreshold stimulation and in agreement with a previous report showing a slight increase of Nur77 expression on CD5hi cells[26]. This validated our in vitro findings and confirmed that tonic TCR signaling and IFN-I signaling jointly upregulate Ly6C. Because the Ly6C$^+$ population is scarce in Ifnar1$^{-/-}$ mice, we have measured CD5 levels on CD8$^+$ Tn cells in these mice and found that compared to WT or IFN-α-treated mice, Ifnar1$^{-/-}$ mice exhibited significantly reduced CD5 levels (Fig. 6H). This implied that tonic TCR signaling is significantly reduced in the absence of type I interferon signaling directly to the T cells. We next measured MHC-I levels on LN stromal cells from wt and IFN-α treated mice. These stromal cells are critically involved in the homeostatic maintenance of Tn cells in the T cell zones of LN[28]. MHC-I levels were significantly increased by IFN-α treatment of wt mice (Fig. 6H). No such increase of MHC-I expression was seen in CD11c+ dendritic cells or T cells (Fig. S4B). Based on this we hypothesize that LN stromal cells are the cell type mediating tonic TCR signals in vivo, however, a formal confirmation of this hypothesis requires further investigation outside of the scope of this study.

**Ly6C$^+$ CD8$^+$ Tn cells are preferentially home to lymph nodes and show improved homeostatic properties.** To investigate homeostatic properties of Ly6C$^+$ cells, we transferred equal numbers of CD8 Tn Ly6C$^+$ and Ly6C$^-$ cells, differentially labeled with Cell Trace Violet (CTV) or CFSE, into lymphopenic RAG-1$^{-/-}$ mice, a situation that leads to vigorous homeostatic proliferation of Tn cells, driven by reduced competition for signals from self-MHC-I[29] and IL-7[30]. Under such conditions Ly6C$^+$ cells proliferated more than their Ly6C$^-$ counterparts and this was more pronounced in the LN (Fig. 7A, complete gating strategy in Fig. S7B) where the number of undivided cells was lower than in the spleen. This suggested that Ly6C$^+$ cells might be exhibiting increased reactivity to IL-7. Indeed, we found that the IL-7R (CD127) exhibited significantly higher expression on Ly6C$^+$ cells (Fig. 7B). Next, we assessed the proliferation of Ly6C-separated CD8$^+$ Tn cell subsets when stimulated with homeostatic cytokines IL-7 and IL-15. Ly6C$^+$, but not Ly6C$^-$, subset proliferated in response to IL-7 (Fig. 7C, left panel), While neither cell type proliferated in response to IL-15 (Fig. 7C, middle panel), the addition of IL-15 did potentiate the proliferative effect of IL-7 specifically in Ly6C$^+$ cells (Fig. 7C, right panel). This suggested that Ly6C$^+$ T cells begin to acquire functional responsiveness to IL-15. To determine if Ly6C$^+$ cells are also more reactive to IL-7 in vivo, we treated mice with IL-7/Anti-IL-7Ab complexes, to increase the in vivo half-life and potency of IL-7. Ly6C$^+$ cells displayed higher reactivity to IL-7c in vivo, as measured by increased expression of the division-associated antigen Ki-67 (Fig. 7D).

Ly6C was previously reported to assist in the LN homing of CM T cells[31]. To assess if this holds for CD8$^+$ Tn cells, we transferred equal numbers of magnetically enriched Tn CD8$^+$s from CD45.1$^+$ control mice (CTV labeled), from IFN-α treated mice (not labeled) and IFN-α treated mice blocked with anti-Ly6C Ab (clone HK1.4; CFSE-labeled). Tn from IFN-α treated mice exhibited higher expression of Ly6C (60.6%) compared to control cells (19.6%). Forty-eight hours after transfer, we identified CD45.1+ donor cells in spleens and LN of recipients and divided them into three populations based on CFSE and CTV. Proportions of all three populations were equal in recipient spleens (Fig. 7E, left panel), but differed in their migration to the LN, where there was an increased proportion of IFN-α treated cells which expressed a higher level of Ly6C, and this increase was completely abrogated by Ly6C blockade (Fig. 7E). These results demonstrate that Ly6C supports the LN homing of Tn cells as well.

Since Ly6C$^+$ cells are BCL-2$^{hi}$ and Eomes$^{hi}$ and induced by IFN-I, we examined the relationship between Ly6C$^+$ phenotype and survival. We incubated magnetically enriched total Tn CD8$^+$s with 100 ng/ml IFN-α either alone or in the presence of CD45$^-$ LN stroma. IFN-α increased survival of Tn CD8$^+$ cells in vitro (Fig. 8A, left panel). Increased survival was even more

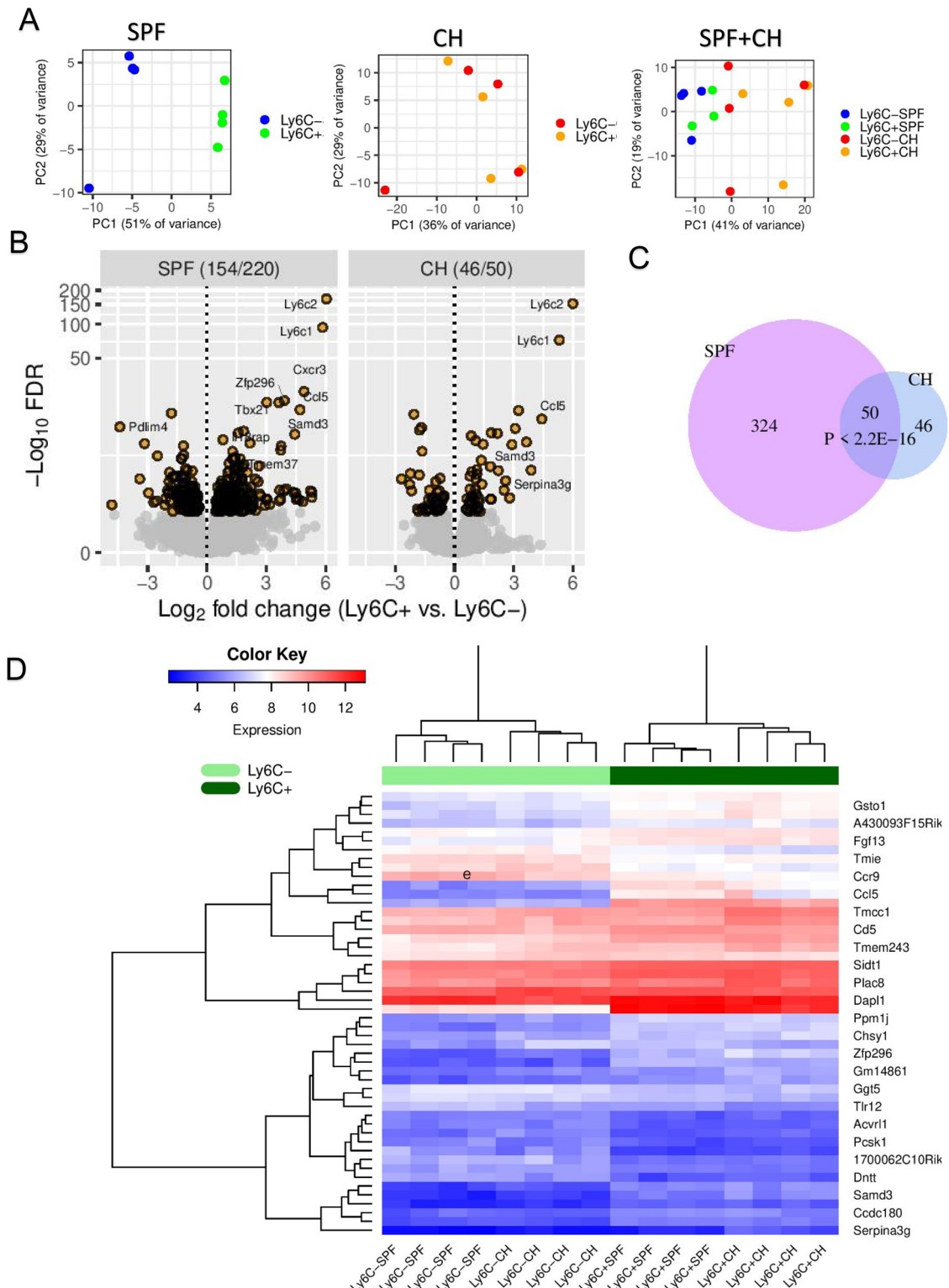

**Fig. 5 Transcriptional distinction between Ly6C$^+$ and Ly6C$^-$ CD8$^+$ Tn cells. A** PCA shows differentially clustered Ly6C$^+$ and Ly6C$^-$ CD8 Tn cells in SPF or co-housed (CH) mice, or both combined (Color scheme: blue-Tn Ly6C$^-$ from SPF mice; green-Tn Ly6C$^+$ from SPF mice; red-Tn Ly6C$^-$ from CH mice; orange-Tn Ly6C$^+$ from CH mice). **B** Volcano plot comparing gene expression changes between Ly6C$^+$ vs. Ly6C$^-$ cells in SPF (left panel) and CH (right panel) mice, where the vertical axis is logarithmically transformed due to very small FDR values of *Ly6C1* and *Ly6C2* genes. Counts of significantly (FDR < 0.05) downregulated and upregulated genes (colored orange), separated by a slash (/), are shown in the top strip label. Genes with log2FoldChange > 3.5 and FDR < 0.0001 are indicated. **C** Venn diagram displaying the significant overlap (*P* < 2.2E−16, Fisher's exact test) between significantly regulated genes in SPF with CH mice. **D** Expression matrix of these 50 genes. Data was normalized through a regularized logarithm transformation (rlog) implemented in DESeq2. Ly6C$^-$ (light green) and Ly6C$^+$ (dark green) cell clusters are highlighted. Data is from one experiment with *n* = 4 mice/group.

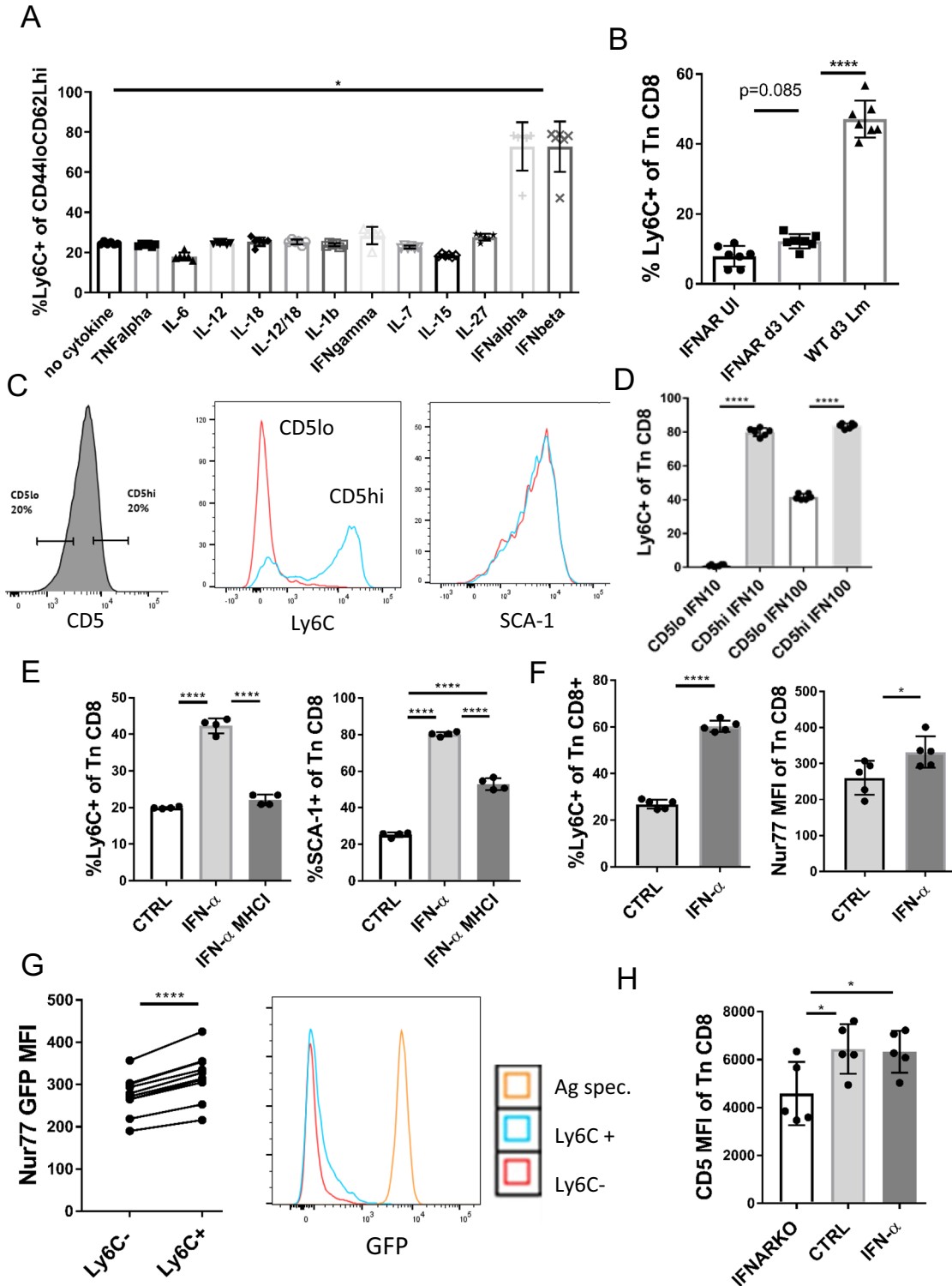

pronounced in coculture with LN stromal cells, where more than half of IFN-α cells survived >7 days (Fig. 8A, right panel). Because LN stromal cells increase survival of Tn cells through the production of homeostatic factors such as IL-7 and CCL21[28], this suggests that IFN-α prolongs Tn survival both directly and in combination with other homeostatic factors. To confirm the role of IFN-I we stimulated Tn CD8$^+$s from Ifnar1$^{-/-}$ and wt mice cocultured with LN stroma from Ifnar1$^{-/-}$ mice. As expected, IFN-α had no effect on the survival of Ifnar1$^{-/-}$ T cells (Fig. 8B, left panel). On the other hand, survival of wt T cells cocultured

with Ifnar1$^{-/-}$ stroma (Fig. 8B, right panel) was increased by IFN-α, similarly to wt stroma. This confirms that type I IFN has a direct homeostatic effect on Tn cells. Simultaneously, we have measured the expression of Ly6C on surviving wt cells cultured with LN stroma with or without IFN-α. CD8$^+$ Tn cells in some wells were labeled with CFSE to determine whether there was any cell proliferation under this condition. By day 7 most (>90%) surviving IFN-α-stimulated cells expressed Ly6C (Fig. 8C), and that was not caused by increased proliferation as CFSE remained undiluted (not shown)., We conclude that this is a consequence of

**Fig. 6 Ly6C upregulation is a direct consequence of IFN-I and tonic TCR signaling. A** CD62L$^{hi}$CD44$^{lo}$ CD8$^+$ T cells were magnetically enriched from pooled secondary lymphoid tissues (spleen, brachial and inguinal lymph nodes) to >95% purity. Cells were then cultured in RPMI media + 10% FCS and 100 ng/ml of each cytokine TNF-α, IFN-γ, IL-12, IL-18, IL-7, IL-15, IL-6, IFN-α, and IFN-β. Twenty-four hours later expression of Ly6C was measured ($n = 6$ mice, data presented as mean ± sd, $p = 0.0237$) **B** Ifnar1$^{-/-}$ mice were infected i.v. with 10$^4$ CFU Lm. Expression of Ly6C on Tn CD8$^+$ cells was measured in Lm infected Ifnar1$^{-/-}$ mice (72 h p.i.), uninfected Ifnar1$^{-/-}$ mice, and wt Lm infected mice ($n = 7$ mice per group, data presented as mean ± sd, ****$p < 0.0001$). **C** Top and bottom 20% of CD8$^+$ Tn cells by CD5 MFI were FACS sorted (Color scheme: red-CD5lo; blue-CD5hi) and (**D**) cultured in RPMI media + 10% FCS and 10 or 100 ng/ml IFN-α. After 24 h expression of Ly6C and SCA-1 were measured by flow cytometry ($n = 6$ mice per group, data presented as mean ± sd, ****$p < 0.0001$). **E** CD8$^+$ Tn cells were magnetically enriched to >95% purity and stimulated with 100 ng/ml IFN-α± MHCI blocking antibody (clone 28-8-6) for 24 h ($n = 4$ mice, mean ± sd). **F** Nur77$^{GFP}$ mice ($n = 5$) were treated with 0.75 μg/mouse of IFN-α. 48 h post-treatment mice were bled retroorbitally and expression of GFP and Ly6C was measured on CD8$^+$ Tn cells ($n = 5$ mice per group, data presented as mean ± sd, ****$p < 0.0001$). **G** Left panel expression of GFP was compared between Ly6C$^+$ and Ly6C$^-$ subpopulations of Tn CD8$^+$ cells ($n = 10$ mice, data presented as individual dots and paired t-test lines, *$p = 0.0363$, ****$p < 0.0001$); right panel GFP histograms of Ly6C ± Tn CD8$^+$ cells were overlayed with GFP histogram of antigen (SIINFEKL peptide, 10$^{-7}$ M) stimulated OT-1 Nur77$^{GFP}$ (Color scheme:red-Ly6C$^-$; blue-Ly6C$^+$, orange-antigen (SIINFEKL peptide) specific). **H** Ifnar1$^{-/-}$ mice, wt mice, and IFN-α treated mice were bled and CD5 MFI was measured on CD8$^+$ Tn cells· ($n = 5$ mice per group, data presented as mean ± sd, left to right *$p = 0.0362$, *$p = 0.0480$). **A**, **B** Data are pooled from two experiments, **C**–**G** Data are representative of two independent experiments (*$p < 0.05$, **$p < 0.01$, ***$p < 0.001$, ****$p < 0.0001$). A one-way Kruskal-Wallis test with Dunn's correction for multiple comparisons, **B**, **D**, **E**, **H** one-way ANOVA with Sidak post hoc correction, **F**, **G** two-tailed Student's t-test.

increased survival and accumulation of Ly6C$^+$ cells. At the same time, surviving Tn cells cultured with stroma alone were gradually losing Ly6C expression in the absence of type I IFN signaling (Fig. 8C). Overall, these results show that IFN-I preferentially increased the survival of Ly6C$^+$ Tn CD8$^+$ cells. Given the preferential LN homing of Ly6C$^+$ cells and the powerful in vitro homeostatic effect of type I IFNs, we have determined absolute numbers of Ly6C+/− subsets of Tn CD8$^+$s in LN of SPF and CH mice. As previously reported, CH mice display a decrease in the proportion of cells with the naïve CD44$^{lo}$CD62L$^{hi}$ phenotype, due to an increase in numbers of memory and effector cells (Fig. S5A). However, because LNs of CH mice exhibited greatly increased overall cellularity (Fig. S5B), they actually exhibit increased absolute numbers of CD8$^+$ Tn cells, which was significant only for the Ly6C$^+$ population (Fig. S5C). Because IFN-I is the main signal for induction of Ly6C on Tn CD8$^+$ cells, we conclude that this accumulation of Ly6C$^+$ Tn is at least partly if not mostly driven by IFN-I signaling.

**Ly6C$^+$ naïve CD8$^+$ T cells show enhanced effector function against low-affinity peptide ligands**. In vitro polyclonal stimulation of CD8$^+$ Tn cells sorted by Ly6C expression showed that Ly6C$^+$ cells produce higher levels of effector molecules, especially GzB. To determine if Ly6C$^+$ CD8$^+$ Tn cells exhibit improved effector function in vivo, we infected B6 SPF mice with Listeria-OVA genetically engineered to express the native immunodominant CD8$^+$ ovalbumin peptide SIINFEKL (N4) or its two altered peptide ligands (APLs) for which the OT-I TCR exhibits decreasing sensitivity: Y3 > Q4[32] (Fig. 9A). Twenty-four hours later we transferred 3 × 10$^4$ sorted Ly6C$^+$ or Ly6C$^-$ cells from OT-1 mice. On day 5 we measured the number of GzB-producing donor cells in the spleen and liver, as well as host bacterial burdens. Mice infected with Listeria expressing native ligand N4 which received Ly6C$^+$ cells exhibited a decreased number of GzB$^+$ cells in the spleen but this was not associated with a lower bacterial burden (Fig. 9B, C). On the other hand, mice infected with Lm expressing lower affinity APL Y3 which received Ly6C$^+$ cells displayed a higher number of GzB producing cells and lower bacterial burden (Fig. 9B, C). A similar trend was observed with the lowest affinity APL Q4—the increase in GzB$^+$ cells in the Ly6C$^+$ group was not statistically significant (Fig. 9B), but the decrease in the bacterial burden was significant (Fig. 9C).

These results indicate that in vivo effector function of Ly6C$^+$ CD8$^+$ Tn cells might be particularly improved for low-affinity clones. To further investigate this, we have treated OT-1 transgenic mice with poly-I:C, isolated Tn CD8s and stimulated them with

APL in vitro. As expected, the majority of OT-1 Tn cells upregulated Ly6C (Fig. S5A). After stimulation with the original agonist N4 (SIINFEKL) peptide, poly-I:C pretreated OT-1 cells exhibited higher production of GzB especially at early time points (24 h and 36 h) while at 72 h untreated cells have almost caught up with GzB production (Fig. 9D-E). A similar trend was observed with all 3 peptides of decreasing affinity (Fig. 9F), indicating that type I interferon exposure is particularly important for rapid acquisition of effector function. Increased IRF4 and Eomes (Fig. 9G) levels, (both transcription factors known to regulate CD8 effector function), pointed to a potential mechanism of how preexposure to type I interferons may increase effector function. Finally, we examined how the poly-I:C pretreatment affects in vivo function by performing transfers (scheme in Fig. S5B, left panel) of Tn CD8$^+$s from untreated and poly-I:C-treated into Lm-OVA APL infected mice. In this in vivo setting, similar to experiments performed with Ly6C$^+$ sorted CD8$^+$ Tn OT-1 cells in vitro, the increased GzB production was evident for the low-affinity APL Q4 (Fig. S5C, left panel). Paralleling that observation, functional reduction of bacterial burden (Fig. S5C, right) was significant and evident for that same low-affinity ligand, providing evidence of functional importance of the increased Ly6C expression on CD8$^+$ Tn cells in the face of low-affinity microbial pathogen variants.

## Discussion

In this work, we uncover a profound impact of bystander infection and inflammation on the CD8$^+$ Tn cell pool in mice housed under 'non SPF conditions'. These conditions induce an expansion of a Ly6C$^+$ subpopulation of CD8$^+$ Tn cells which are Eomes$^{hi}$BCL-2$^{hi}$CD5$^{hi}$SCA1$^{hi}$ and exhibit rapid effector function upon TCR stimulation. The Human Ly-6/uPAR protein family has 20 members, yet has no known human homolog of Ly6C[33]. The exact function of Ly6C is unknown but it has been reported that Ly6C assists LN homing of central Tm cells[31]. Our transfer experiments support that this is the case for CD8$^+$ Tn cells and that increasing Ly6C expression leads to superior LN homing.

Our in vitro data shows that IFN-I can selectively and specifically upregulate Ly6C on CD8$^+$ Tn cells, whereas CD8$^+$ Tn cells deficient for the IFN-I receptor expressed very little Ly6C. That allowed us to identify IFN-I as the main signals that upregulate Ly6C in a bystander manner on CD8$^+$ Tn cells. While this upregulation was transient in viral and bacterial monoinfections, it was permanent under 'non-SPF' conditions.

IFN-I are known for their role as signal 3 cytokines, involved in directing effector immunity, as well as T cell memory formation[34,35]. Homeostatic roles of IFN-I on T cells are much less

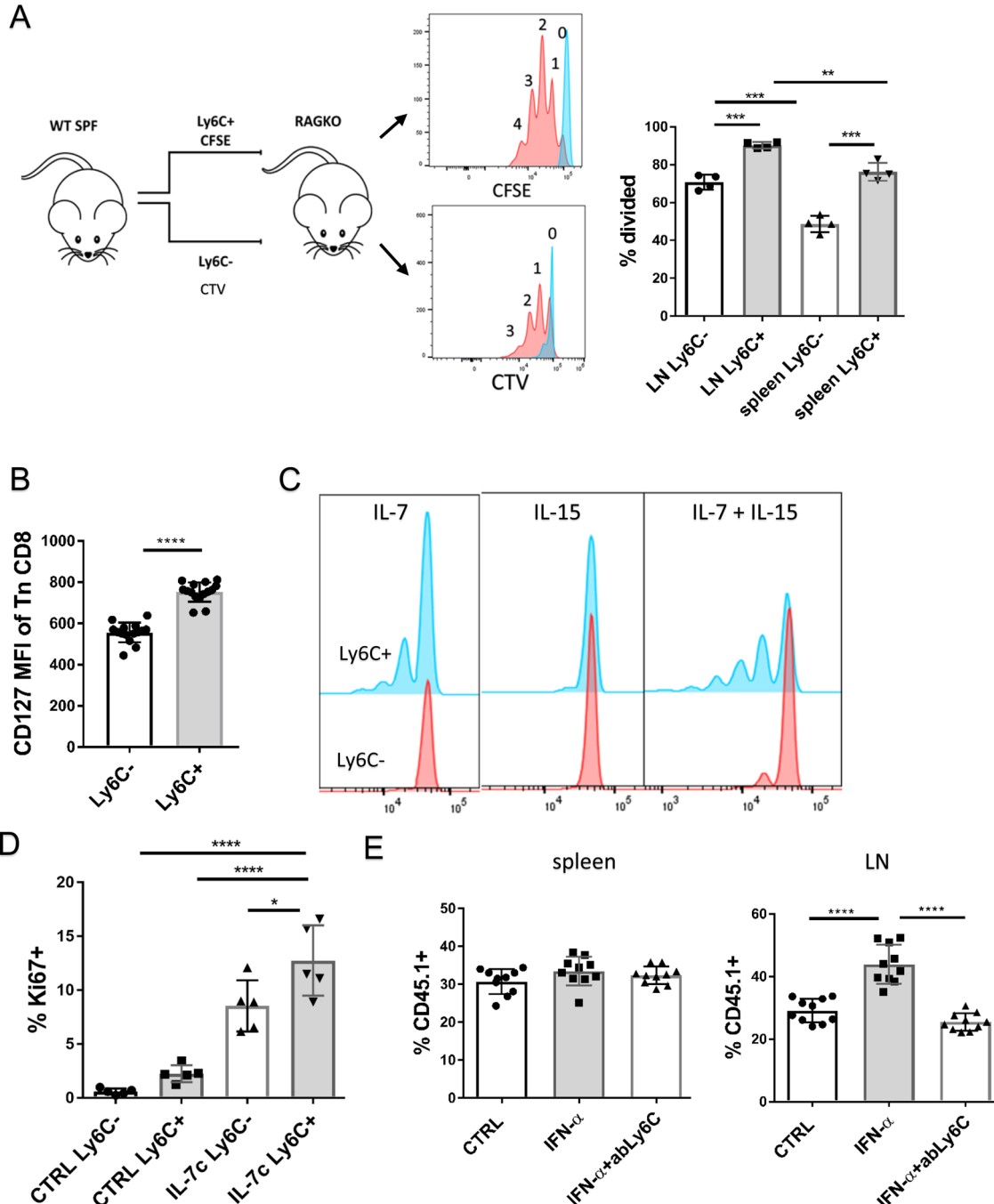

**Fig. 7 Ly6C$^+$ CD8$^+$ Tn cells preferentially home to lymph nodes and show improved homeostatic properties. A** CD8$^+$ T cells were magnetically enriched isolated LN and spleen of C57BL/6 mice ($N = 6$), Ly6C$^+$ and Ly6C$^-$ Tn cells were sorted by FACS. Ly6C$^+$ cells were labeled with CFSE and Ly6C$^-$ with CTV and $4 \times 10^5$ cells were adoptively transferred into RAG-1$^{-/-}$ hosts Four days later, the proliferation of donor cells was analyzed by FACS (Color schemed: blue-undivided cells; red-divided cells) in the LN and spleen of host mice ($n = 4$ recipient mice, data presented as mean ± sd, $^*p = 0.001$, $^{***}p < 0.001$). **B** MFI of IL-7R (CD127) was measured on CD8$^+$ Tn cells from C57/BL6 mice ($n = 15$ mice, data presented as mean ± sd). **C** CD62L$^{hi}$CD44$^{lo}$ CD8$^+$ T cells were magnetically enriched from pooled secondary lymphoid tissues ($n = 5$ mice) and FACS sorted by Ly6C expression. Ly6C ± cells were labeled with CFSE and stimulated with 500 ng/ml rIL-7, 100 ng/mL rIL-15, or both. 7 days letter proliferation was measured on live cells (Color scheme: red-Ly6C$^-$ cells; blue-Ly6C$^+$ cells). **D** C57BL/6 mice were treated with IL-7/Anti-IL-7Ab (M25) complexes (1.5 µg/mouse every 48 h). On day 6 after the start of treatment mice were bled retroorbitally and expression of Ki67 was measured on CD8$^+$ Tn cells ($n = 5$ per group, data presented as mean ± sd, $^*p = 0.02$, $^{****}p < 0.0001$). **E** We transferred $2 \times 10^6$ of magnetically enriched Tn CD8$^+$s from CD45.1 control mice (CTV labeled), from IFN-$\alpha$ treated mice (48 h prior to harvest; not labeled) and IFN-$\alpha$ treated mice blocked with anti-Ly6C Ab (clone HK1.4; CFSE-labeled) to 5 recipient C57BL/6 mice. 20 min later spleen and LN pools (inguinal, brachial, cervical) were harvested and the proportion of three transferred cell populations was measured ($n = 10$ recipient mice, data presented as mean ± sd). **A, C, D** data are representative of two independent experiments. **B, E** data are pooled from two experiments ($^*p < 0.05$, $^{**}p < 0.01$, $^{***}p < 0.001$, $^{****}p < 0.0001$). **A, D, E**, one-way ANOVA, with Sidak post hoc correction; **B** paired two-tailed Student's test.

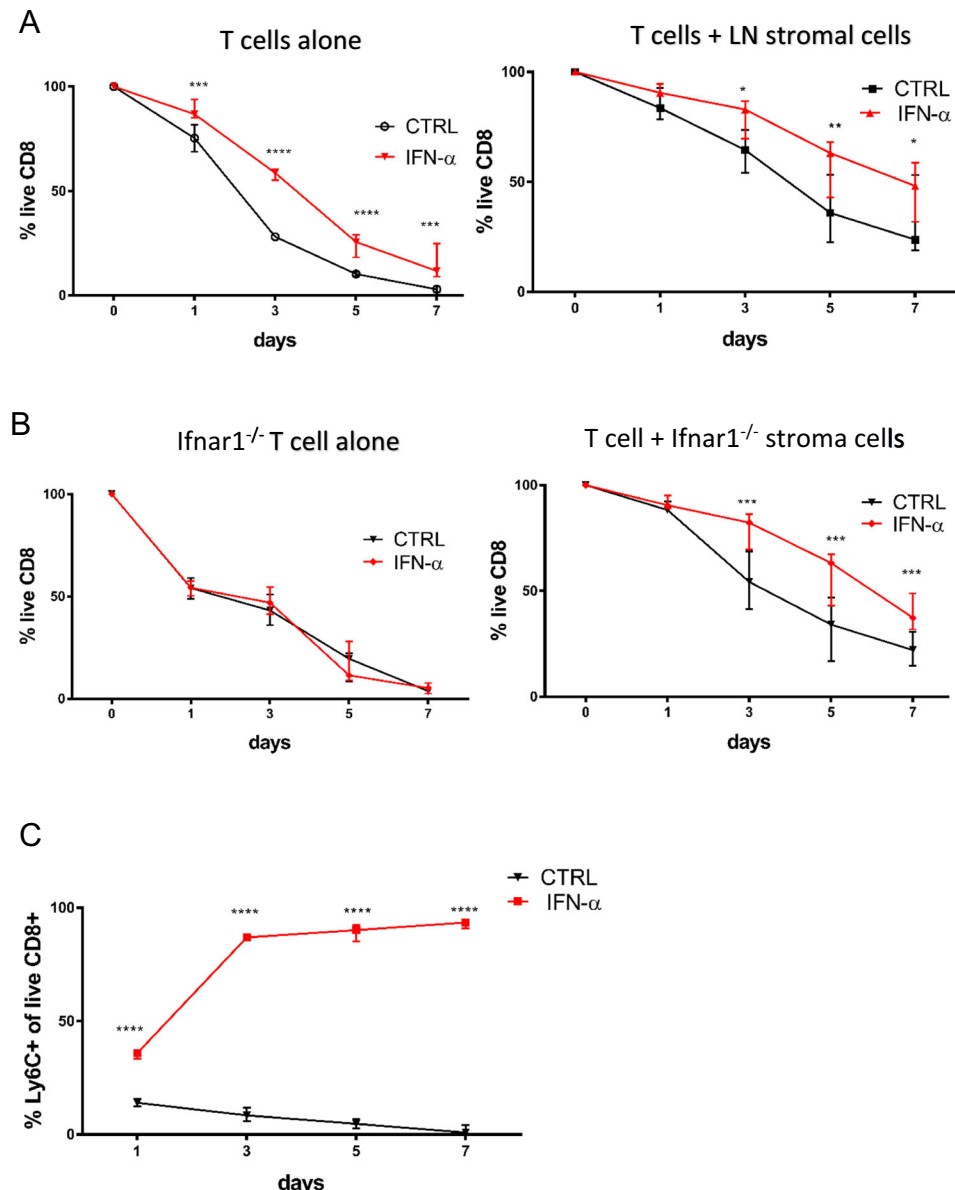

**Fig. 8 Type I interferons enhance survival of preferentially Ly6C+ CD8+ Tn cells in vitro. A** We magnetically enriched CD8+ Tn cells to >95% purity. $3 \times 10^5$ cells were cultured in round bottom 96 well plates either alone or in the presence CD45- LN stroma in RPMI media + 10% FCS and 0.05 mM beta-mercaptoethanol ±100 in ng/ml IFN-α. Cell viability was assessed by flow cytometry on days 1, 3, 5, and 7 of culture ($n = 8$ mice, data presented as mean ± sd). **B** viability of wt CD8+ Tn cells cocultured with Ifnar1$^{-/-}$ stroma and vice versa Ifnar1$^{-/-}$ CD8+ Tn cells stimulated by 100 ng/ml IFN-α was measured on days 1, 3, 5, and 7 ($n = 8$ mice, data presented as mean ± sd). **C** expression of Ly6C on live CD8+ Tn wt cells cultured with LN stroma with or without IFN-α ($n = 8$ mice, data presented as mean ± sd). Data are pooled from two experiments, $n = 8$, (*$p < 0.05$, **$p < 0.01$, ***$p < 0.001$, ****$p < 0.0001$). **A**, **B**, **C** For the survival of cultured T cells cultured with or without stroma, statistical significance was calculated with a paired two-tailed Student's *t*-test between corresponding time points.

studied[36], with the exception that it has been shown that they induce bystander proliferation of memory T cells[10] and subsequent attrition[37] during viral infection. Even before antigen presentation, exposure to IFN-I sensitizes T cells to rapidly exert effector functions, such as IFN-production, on contact with their cognate antigen[38]. Our results show that IFN sensitization of CD8+ Tn cells programs such cells to express Ly6C and exhibit rapid and enhanced production of effector molecules. This was especially pronounced for low-affinity clones, consistent with previous data[39]. This effect is likely mediated by IFN-I induction of the Eomes transcription factor, which is necessary for the expression of effector molecules and memory formation[21].

Further clues into the biology of Ly6C+ cells were provided by their high expression of CD5 and BCL-2. CD5 is a negative regulator of TCR signaling and its expression is considered a measure of the strength of encounter with self-ligand[26]. CD5$^{lo}$ cells were not able to upregulate Ly6C when stimulated with IFN-I. Moreover, blocking of MHCI molecules abrogated Ly6C upregulation. This showed that tonic TCR signaling is a necessary cofactor for IFN-I induced Ly6C expression. It is well established that Tn homeostasis depends on weak, tonic signaling from the TCR:spMHC interaction[40] followed by IL-7 signaling[30]. IL-7-mediated survival is regulated by the transcription factors BCL-2 and MCL-1. Therefore, increased BCL-2 and MCL-1 expression on Ly6C+ Tn cells suggested increased IL-7

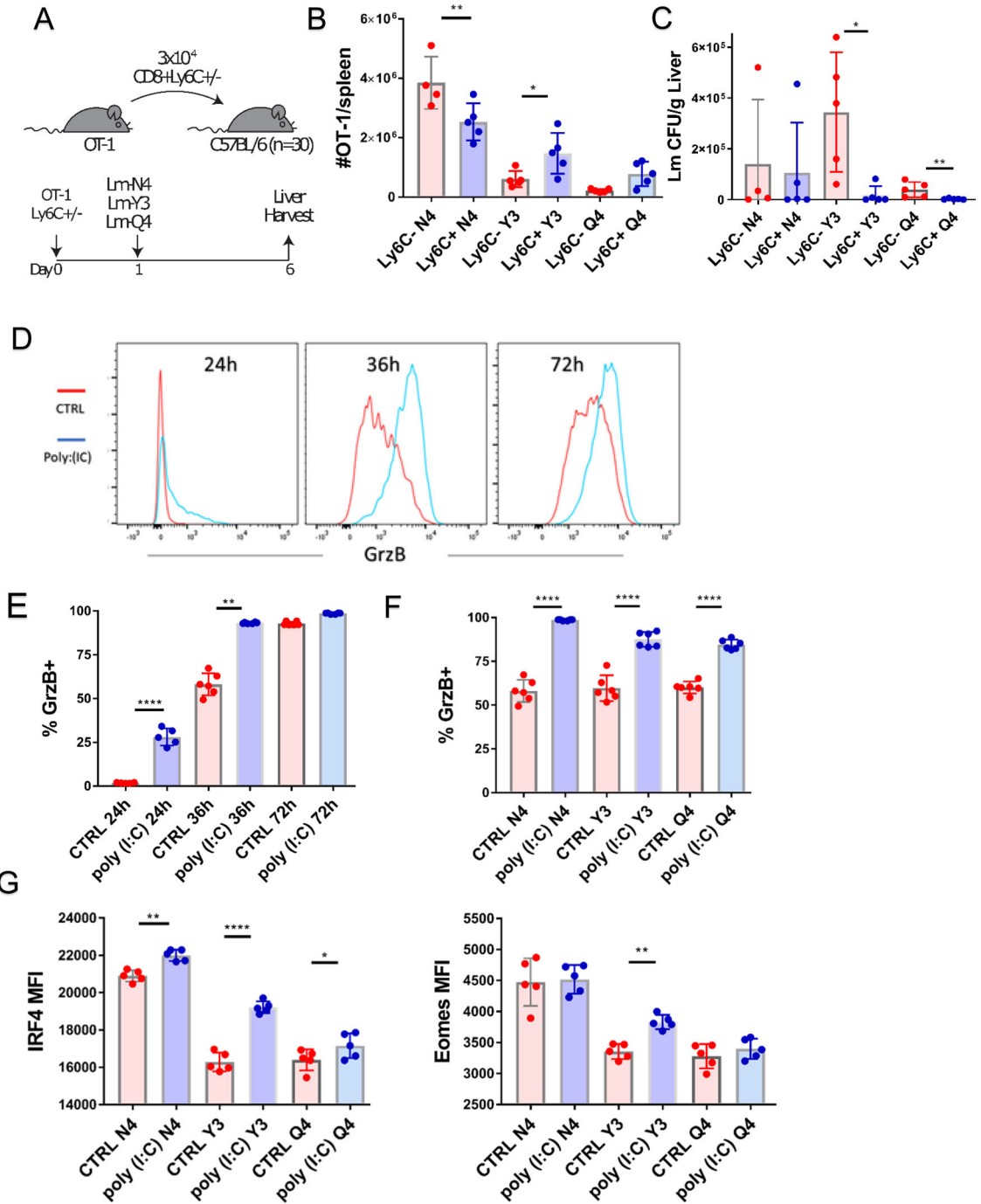

**Fig. 9 Ly6C+ naïve CD8+ T cells show enhanced effector function against low-affinity alternative peptide ligands. A** Mice were inoculated i.v. with ~10⁴ CFU Lm genetically engineered to express the native immunodominant CD8+ ovalbumin peptide SIINFEKL (N4) or its two altered peptide ligands (APLs) Y3 and Q4. Twenty-four hours later we transferred $3 \times 10^4$ sorted Ly6C± cells from OT-1 mice into total 6 groups of mice ($n = 5$ mice per group) (transfer scheme). **B** On day 5 post-transfer the number of GzB-producing donor cells in the spleen and liver ($n = 5$ mice per group, data presented as mean ± sd, *$p = 0.0372$, **$p = 0.0029$). **C** Host bacterial burdens in the liver were measured for every Lm APL administered ($n = 5$ mice per group, data presented as mean ± sd, *$p = 0.0159$, **$p = 0.0079$). **D**, **E** Tn CD8+s were magnetically enriched from control and poly(I:C) treated mice (Color scheme: red-control mice; blue-poly(I:C) treated mice) and stimulated in vitro with $10^{-7}$ M SIINFEKL peptide and Y3 and Q4 APLs. Production of GzB was measured at 24 h ($n = 5$ mice), 36 and 72 h of stimulation ($n = 6$ mice, data presented as mean ± sd, *$p = 0.0022$, ****$p < 0.0001$) **F** At 36 h GzB expression was higher in cells from mice pretreated with poly(I:C) when stimulated with any of the three peptides ($n = 6$ mice, data presented as mean ± sd). **G** At the same time point expression of transcription factors Eomes and IRF4 was increased in poly(I:C) pretreated cells ($n = 5$ mice, data presented as mean ± sd, *$p = 0.0491$, **$p = 0.0031$, ****$p < 0.0001$). **A–G** Data are representative of two independent experiments (*$p < 0.05$, **$p < 0.01$, ***$p < 0.001$, ****$p < 0.0001$). **B** $p$ values calculated by unpaired Students $t$-test between mice groups receiving same APL, **C** Mann-Whitney U test between mice groups receiving same APL **E**, **F**, **G** Students $t$-test.

reactivity. Indeed, Ly6C$^+$ cells expressed higher levels of the IL-7R and proliferated more when stimulated with IL-7 in vitro. Consistent with that, Ly6C$^+$ cells also proliferated more vigorously in the lymphopenic RAG-KO environment, especially in the LN.

In addition, IFN-I directly increased the survival of CD8$^+$ Tn cells in culture, with the Ly6C$^+$ cells surviving the longest. Furthermore, in vivo treatment of Nur77-driven GFP reporter mice[27] with type I IFNs resulted in low but reproducible increased GFP expression, providing evidence for increased homeostatic TCR signaling. Thus, our results point to IFN-I as the new homeostatic factor, which maintains the Tn Ly6C$^+$ subpopulation characterized by increased lymph node homing, tonic TCR signaling, and IL-7 reactivity.

Lastly, we show that Ly6C$^+$CD8$^+$ Tn cells exhibit increased effector function particularly towards low-affinity APLs in vivo. This suggests that Ly6C$^+$ cells might be recruited in the response at a higher rate when ligands are of lower affinity, and this may result in their increased lymph node or tissue homing. Contrary to in vivo data where effector function was improved only with low-affinity ligands our in vitro data showed increased effector function with all three APLS. The reason for this discrepancy could be immediate exposure of transferred cells to host type I IFNs which is difficult to circumvent as IFNARKO Tn CD8$^+$s do not express Ly6C and cytokine knockouts do not exist. These findings might have practical implications for adoptive T cell therapy of tumors, which will require further investigation.

## Methods

**Ethics statement**. Mouse studies were carried out in strict accordance with the recommendations in the Guide for the Care and Use of Laboratory Animals of the National Institutes of Health. Protocols were approved by the Institutional Animal Care and Use Committee at the University of Arizona (IACUC protocol 08-102, PHS Assurance No. A3248-01).

**Mice**. Female C57/BL6 mice and OT-I TCR transgenic mice were obtained from Jackson Laboratories (Bar Harbor, ME). IFNAR-KO mice (IFN-αβR-) were bred in our colony from the nucleus obtained from Jackson Laboratories. Adult C57BL/6 (B6) mice were infected with 1000 plaque-forming units of the West Nile virus strain 385-99[8] via footpad injection.

For co-housing, we purchased female outbred pet-store mice from two local vendors, individually tagged them, and co-housed them for two weeks to maximize microbial exchange in large (rat) cages. We then co-housed them in large cages with young female C57BL6 mice (Jackson Laboratories) separated by a perforated barrier. All mice were bled before co-housing (baseline) and then at 30 and 60 days, followed by the sacrifice of certain animals on post day 60. We have selected a 60 day period as the first harvest point since this is a conventionally accepted time for any acute immune response to entering the memory phase steady state[41].

**Flow cytometry and cytokine measurements**. Blood was collected into heparin-coated tubes and lysed hypnotically. Splenocytes were mechanically dissociated through a 40 μm plastic mesh to prepare a single-cell suspension. Cells were stained with surface antibodies, and then fixed and permeabilized using the FoxP3 Fix/Perm kit (eBioscience, San Diego, CA). Following antibodies were used anti: Ly6C(HK1.4), CD3(17A2), CD8a(53-6.7), CD44(IM7), CD4 (GK1.5), CD5 (53-7.3), CD62L(MEL-14), CD25(PC6.1), SCA1(D7), CD69(H1.2F3), CXCR3 (CXCR#173) (Biolegend, San Diego, CA), Eomes(Dan11mag), Phospho-ZAP70/Syk(n3kobu5) (Invitrogen, Carlsbad, CA). Cytokines were measured with bead-based flow cytometric immunoassay (Legendplex, Biolegend). Serum levels of IFN-I were measured using a bioassay as previously described[8]. Type I IFN standards (National Institute of Allergy and Infectious Diseases international standard) and mouse serum were serially diluted twofold in complete media, 10% FBS, Pen/Strep, and DME. IFN-responsive L929 cells were plated at $5 \times 10^4$ cells/well, incubated overnight with the serum, and media containing 5 PFU VSV Indiana was added to each well, except for control wells. Twenty-four hours later, media was aspirated, the plate washed twice with PBS, and the monolayer fixed with 5% formaldehyde, incubated for 10 min, and stained with 0.05% crystal violet for 10 min. Washed monolayers were allowed to dry, and 100% methanol was used to elute the dye. Absorbance is measured at 595 nm on an ELISA plate reader (Invitrogen).

Samples were acquired using a BD LSR Fortessa cytometer with FACS DIVA v 8.0.1. software (BD Biosciences) and analyzed by FlowJo software (Tree Star, Ashland, OR) with a minimum of 20000 CD8$^+$ T cells collected per sample. For MHC-I blocking anti-H-2Kb/H-2Db antibody (clone 28-8-6, Biolegend) was used. Phosphorylation of Erk and ZAP-70 was measured using Phosflow fixation and permeabilization solutions (BD Biosciences, San Jose, CA) according to the manufacturer's instructions.

Flow cytometry files from SPF and WNV infected mice were uploaded to Cytobank, a cloud-based computational platform. Dead cells were excluded by manual gating and CD3+ events were input into the FLOWSOM clustering algorithm.

**Cell separation by MACS enrichment and in vitro stimulation of naïve T cells**. Naïve splenic CD8$^+$ T cells were magnetically enriched using an AutoMACS pro (Miltenyi Biotec) using the CD8a T cell isolation kit supplemented with anti-CD44-biotin (eBioscience). The cells were cultured at $2 \times 10^5$ cells/ml in RPMI-1640 with L-glutamine (Lonza, Basel, Switzerland) + 10% FCS 1:1 with α-CD3/α-CD28 immobilized on beads (Miltenyi), 10 U/ml rmIL-2 (eBioscience). The purity of isolated naïve T cells was assessed by flow cytometry and was >95% in all experiments. Lymph node (LN) stromal cells were obtained by dissociating lymph nodes in Liberase TL enzyme mixture (Roche, Basel, Switzerland) using Gentle-MACS tissue dissociator (Miltenyi Biotec, Bergisch Gladbach, Germany). When lymph nodes were fully digested lymph node stromal cells were negatively enriched using anti CD45 and anti Ter119 magnetic beads (Miltenyi Biotec). Stromal cells were passaged 5–10 times in alpha-modified MEM (Sigmaaldrich, St.Louis, Mo) with 20% FCS. Naïve T cells were cultured with lymph node stromal cells in 1:10 ratio in 96 well-round plates.

**Adoptive transfer of naïve T cells**. Adoptive transfers of naïve CD8$^+$ T cells were performed as previously described[42]. Briefly, naïve CD8$^+$ T cells from TCR transgenic OT-1 mice were sorted by Ly6C expression on the FACS Aria III sorter (BD Biosciences) based on the phenotype of CD8$^+$C62L$^+$CD44$^{lo.}$ CD45.1$^+$OT-1 Ly6C$^+$ or Ly6C$^-$ cells were transferred into adult congenic B6 recipients; 24 h later, recipient mice were infected with $1–3 \times 10^4$ CFU of Lm expressing or not expressing the immunodominant epitope of ovalbumin (SIINFEKL, OVA)[43] (Lm-OVA) intravenously, in 100 μl PBS. Analysis was performed on days 5 and 8 after infection.

**RNAseq analysis**. $2 \times 10^5$ CD8$^+$CD62L$^+$CD44$^{LO}$Ly6C$^+$and Ly6C$^-$ naïve T cells were isolated from 4 SPF and 4 cohoused mice using a CD8a$^+$ T cell isolation kit (Miltenyi Biotec) followed by FACS Aria III sorting (BD Biosciences). Raw sequence reads were first demultiplexed using the SMART-Seq DE3 Demultiplexer software (www.takarabio.com/products/next-generation-sequencing/rna-seq/ultra-low-input-rna-seq/smart-seq-de3-demultiplexer) and read 1 file were used. Sequencing reads were evaluated using FastQC (v0.11.3, www.bioinformatics.babraham.ac.uk/projects/fastqc/) and preprocessed for quality using Trim Galore (v0.4.0, Phred score threshold of 20 and minimum length of 50 bp, www.bioinformatics.babraham.ac.uk/projects/trim_galore/). Quality reads were then uniquely mapped to the mouse reference genome (GRCm38) using Tophat2 (v2.1.1). Raw read counts for each sample were obtained by mapping reads at the gene level using HTSeq-count tool from the Python package HTSeq. DESeq2 R package (v1.8.2) was then used to perform differential expression and statistical analysis. Functional enrichment of the differentially expressed genes relative to a background gene list (from all filtered expressed genes) was performed using clusterProfiler R package.

**Statistical analysis**. SPSS and Graph Pad Prism were used for statistical analysis. Depending on data distribution, differences were calculated by Student's $t$-test, Mann Whitney U-test, one-way ANOVA, or Kruskal Wallis test with SIdak post hoc correction. For all statistical differences $*p < 0.05$, $**p < 0.01$, $***p < 0.001$, $****p < 0.0001$.

**Reporting summary**. Further information on research design is available in the Nature Research Reporting Summary linked to this article.

## Data availability

The data and materials that support the findings of this study are available from the corresponding author upon request. Source data are provided with this paper. The RNA-seq data discussed in Fig. 5 of this publication have been deposited in NCBI's Gene Expression Omnibus and are accessible through GEO Series accession number GSE159076. The flow cytometric data used for unbiased clustering in Fig. 1 has been deposited to flow repository.org and is accessible through accession number FR-FCM-Z4AS. Source data are provided with this paper.

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

## Acknowledgements

The authors would like to thank University of Arizona Flow Cytometry core facility personnel for help with cell sorting. We would also like to thank all the members of the JNZ lab for constructive discussion. Supported by USPHS awards AG020719 and AG048021 and the Elizabeth Bowman Endowed Chair in Medical Sciences to J.N-Z.

## Author contributions

M.J., M.S., D.B. and J.N.-Z. designed experiments. M.J., C.P.C., J.L.U. and S.C. performed experiments. M.J., C.P.C., D.B., J.L.U. and S.C. analyzed the data. M.J. and J.N.-Z. wrote the manuscript. J.N.-Z. administered the project.

## Competing interests

J.N.Ž. is on the scientific advisory board and receives research funding from Young Blood, Inc. The remaining authors declare no competing interests.
