## [Peer Review File · Nature Communications]

REVIEWERS' COMMENTS

Reviewer #1 (Remarks to the Author):

I think the authors did a good job of responding to Reviewer #1s concerns, and it looks like they were attentive to the other reviewers' comments too. I did read through the manuscript and overall I'd say it's interesting and important, with the major points being quite well demonstrated.

If I were reviewing it for the first time I'd have two main concerns -- the unclear Figure 1 that I already mentioned to you, and the finding that OT-I cell control of recombinant LM infection seemed to get worse as the TCR affinity increases (Fig. 9C -- very counterintuitive and needing further discussion). However, I feel it unreasonable to bring up new concerns at this stage, especially if I'm serving as a moderator. So, in my opinion, the revised manuscript has sufficiently addressed previous concerns and I'd vote for acceptance for publication.

Reviewer #2 (Remarks to the Author):

The authors resolved all remaining points.

Reviewer #3 (Remarks to the Author):

The authors have adequately answers all questions and criticisms.

REVIEWERS' COMMENTS

Reviewer #1 (Remarks to the Author):

I think the authors did a good job of responding to Reviewer #1's concerns, and it looks like they were attentive to the other reviewers' comments too. I did read through the manuscript and overall I'd say it's interesting and important, with the major points being quite well demonstrated.

If I were reviewing it for the first time I'd have two main concerns -- the unclear Figure 1 that I already mentioned to you, and the finding that OT-I cell control of recombinant LM infection seemed to get worse as the TCR affinity increases (Fig. 9C -- very counterintuitive and needing further discussion). However, I feel it unreasonable to bring up new concerns at this stage, especially if I'm serving as a moderator. So, in my opinion, the revised manuscript has sufficiently addressed previous concerns and I'd vote for acceptance for publication.

We thank the reviewer for the comments.

We have added the following sentence to further clarify Figure 1:

"FlowSOM software visualizes flow cytometric data as a minimal spanning tree result very similar to more commonly used SPADE software but with greater computing speed."

As Figure 9C is concerned we would like to point out that this finding on decreased GzB positive cells *in vivo* was not associated with lower bacterial burden or decreased function with N4 APL *in vitro* so the conclusion that Lm infection control decreases with higher TCR affinity would be far fetched. To further clarify reviewer's concern we have reworded the result section describing Figure 9C to state this:

"Mice infected with Listeria expressing native ligand N4 which received Ly6C+ cells exhibited decreased number of GzB+ cells in the spleen but this was not associated with a lower bacterial burden (Figure 9 B,C)."

Also, we have rephrased the final part of the Discussion section to state this:

'Contrary to in vivo data where effector function was improved only with low affinity ligands our in vitro data which showed increased effector function with all three APLs. The reason for this discrepancy could be immediate exposure of transferred cells to host type I IFNs which is difficult to circumvent as IFNAR1 KO Tn CD8+ do not express Ly6C and cytokine knock outs do not exist.'

We hope that this will successfully address reviewer's final concern.

Reviewer #2 (Remarks to the Author):

The authors resolved all remaining points.

We thank the reviewer for their comments.

Reviewer #3 (Remarks to the Author):

The authors have adequately answers all questions and criticisms.

We thank the reviewer for their comments.